# Application of *In vitro* transcytosis models to brain targeted biologics

**Kangwen Deng**[1]*, **Yifeng Lu**[1], **Sjoerd J. Finnema**[2], **Kostika Vangjeli**[1], **Junwei Huang**[1], **Lili Huang**[1], **Andrew Goodearl**[1]

**1** AbbVie Bioresearch Center, Worcester, MA, United States of America, **2** AbbVie Inc., North Chicago, IL, United States of America

* Katherine.deng@abbvie.com

**Data Availability Statement:** All relevant data are within the manuscript and its Supporting Information files.

**Funding:** All authors (K.D., Y.L., S.J.F., K.V., J.H., L.H., and A.G.) are employees of AbbVie. This

## Abstract

The blood brain barrier (BBB) efficiently limits the penetration of biologics drugs from blood to brain. Establishment of an *in vitro* BBB model can facilitate screening of central nervous system (CNS) drug candidates and accelerate CNS drug development. Despite many established *in vitro* models, their application to biologics drug selection has been limited. Here, we report the evaluation of *in vitro* transcytosis of anti-human transferrin receptor (TfR) antibodies across human, cynomolgus and mouse species. We first evaluated human models including human cerebral microvascular endothelial cell line hCMEC/D3 and human colon epithelial cell line Caco-2 models. hCMEC/D3 model displayed low trans-epithelial electrical resistance (TEER), strong paracellular transport, and similar transcytosis of anti-TfR and control antibodies. In contrast, the Caco-2 model displayed high TEER value and low paracellular transport. Anti-hTfR antibodies demonstrated up to 70-fold better transcytosis compared to control IgG. Transcytosis of anti-hTfR.B1 antibody in Caco-2 model was dose-dependent and saturated at 3 µg/mL. Enhanced transcytosis of anti-hTfR.B1 was also observed in a monkey brain endothelial cell based (MBT) model. Importantly, anti-hTfR.B1 showed relatively high brain radioactivity concentration in a non-human primate positron emission tomography study indicating that the *in vitro* transcytosis from both Caco-2 and MBT models aligns with *in vivo* brain exposure. Typically, brain exposure of CNS targeted biologics is evaluated in mice. However, antibodies, such as the anti-human TfR antibodies, do not cross-react with the mouse target. Therefore, validation of a mouse *in vitro* transcytosis model is needed to better understand the *in vitro in vivo* correlation. Here, we performed transcytosis of anti-mouse TfR antibodies in mouse brain endothelial cell-based models, bEnd3 and the murine intestinal epithelial cell line mIEC. There is a good correlation between *in vitro* transcytosis of anti-mTfR antibodies and bispecifics in mIEC model and their mouse brain uptake. These data strengthen our confidence in the predictive power of the *in vitro* transcytosis models. Both mouse and human *in vitro* models will serve as important screening assays for brain targeted biologics selection in CNS drug development.

study was fully funded by AbbVie. AbbVie participated in design and conduct of experiments, interpretation of data, and review and approval of the manuscript.

**Competing interests:** The authors have declared that no competing interests exist.

**Abbreviations:** BBB, Blood-brain barriers; bEnd3, murine brain endothelial cell; Caco-2, Human Caucasian Colon Adenocarcinoma Epithelial Cells; CNS, central nervous system; DVD-Ig, Dual-variable-domain Immunoglobulin; hCMEC/D3, immortalized human capillary endothelial cell; LLOQ, lower limit of quantitation; MBT, monkey (Macaca irus) brain capillary endothelial cells, brain pericytes and astrocytes. BBB Kit; mIEC, murine intestinal epithelial cells (CI-muADINTESTI); MSD, Electrochemiluminescence-Meso Scale Discovery; NHP, nonhuman primate; p-SCN-Bn-Deferoxamine, 1-(4-isothiocyanatophenyl)-3-[6,17-dihydroxy-7,10,18.21-tetraoxo-27-(N-acetylhydroxylamino)-6,11,17,22-tetraazheptaeicosine]thiourea; PET, Positron emission tomography; RGMa, Repulsive guidance molecule a; RMT, receptor-mediated transcytosis; SUV, standardized uptake value; TfR, transferrin receptor 1.

# Introduction

Due to the global increase in age-related central nervous system (CNS) diseases, there is substantial demand for new CNS medicines. Biologics has become the new trend for CNS drug discovery. Despite the large number of candidates in the pipeline and clinical trials, the success rate of CNS biologics drug is extremely low [1]. Low blood–brain barrier (BBB) permeability has been one of the major causes of failure for new CNS drug candidates. The BBB is formed by specialized brain microvascular endothelial cells (BMECs) and other supporting cells of the neurovascular unit including pericytes, astrocytes, and neurons [2]. The low permeability is caused by decreased transcytotic activity and by tight junctions and adherents' junctions that limit paracellular passage [3]. BBB plays a key role as a critical protecting barrier for the CNS against toxic and infectious agents while maintaining the ionic and volumetric environments. The barrier properties also create an obstacle for effective systemic drug delivery to the CNS from blood in which 0.01–0.4% percent of all large molecule drugs gets access [4]. Therefore, there has been a great interest in cell models which mimic BBB permeation properties for the purposes of drug screening and engineering [5, 6].

Receptor-mediated transcytosis (RMT) has been proven as a feasible approach for enhancing protein drug delivery to brain [7–10]. Specific BBB receptors such as transferrin receptor (TfR), insulin receptor, and insulin-like growth factor 1 receptor (IGF1R) are triggered by ligand binding to internalize and traffic through brain endothelial cells (BEC), and in some cases to release ligand cargo on the abluminal surface of the BBB [11–13]. Targeting these endogenous mechanisms of transcytosis is considered one of the most promising approaches to develop a non-invasive, safe, and specific cross-BBB delivery of biologics [14, 15]. The RMT delivery strategy is based on using the coupling of a non-transportable therapeutic protein to a transportable peptide or protein, which undergoes receptor-mediated transcytosis through the BBB. TfR is the best-known BBB target. Currently, there are approved drug [16] and clinical stage biologics [10] utilizing TfR targeted fusion protein approach to enhance brain uptake of therapeutic biologics through the RMT mechanism.

Affinity and species reactivity characteristic of antibody require the development of species-specific *in vitro* transcytosis model. Models with human origin are more relevant for screening of human BBB targeted biologics. Models from primary human brain capillary endothelial cells (BCEC) [17–20] and immortalized cells, such as human cerebral microvascular endothelial cell line hCMEC/D3 [21], have been used in some studies despite their high paracellular permeabilities. More recently, human induced pluripotent stem cell (iPSC)-derived BBB models with high trans-epithelial electrical resistance (TEER) value have been developed [22, 23]. Also, more sophisticated models with sheer stress and multi-cellular support are reported [24, 25]. These models are validated by endothelial phenotype, expression of tight junction proteins and small molecule transport [26]. Given the lack of human brain exposure data of biologics, the applications of these models to biologics screening have not been validated. Therefore, even though many *in vitro* human BBB models exist, the prediction power of these models for brain penetration to biologics remains unknown.

At early stages of drug development, *in vivo* studies are mostly performed in rodents. Rodent brain exposure data of biologics are available [8, 9, 14, 27] and validation data can be readily generated. Rat primary or immortalized BEC-based model, such as rat RBE4 [27–30] have been used to some extent for BBB-targeted antibody selection. The lack of sufficient barrier properties of the immortalized mouse brain endothelial cell line bEND3 cells [31], has hindered its potential as an accurate model. The utility of rodent *in vitro* BBB models in biologics screening and *in vitro in vivo* correlation has not been fully evaluated.

To better understand the utility of these *in vitro* BBB models for screening of biologics, we have performed systematic transcytosis of anti-TfR antibodies and / or bispecifics in mouse cells, monkey cells and human cells models. Our data indicates that high TEER value is

required for a successful transcytosis model. Epithelial cell models, such as the murine epithelial cell line mIEC or the human colorectal adenocarcinoma cell line Caco-2, can be useful for biologics screening. We showed a good correlation between *in vitro* transcytosis of anti-mouse TfR molecules and their *in vivo* brain uptake in mice. Good differentiation of anti-human TfR antibodies was also observed in Caco-2-based transcytosis assay.

Anti-hTfR.B1 has comparable binding potency to human and cynomolgus monkey TfR. *In vitro* transcytosis of anti-hTfR.B1 antibody showed better transcytosis in both Caco-2 and monkey brain transcytosis (MBT) models compared to control antibody. Enhanced brain radioactivity concentration of radiolabeled anti-hTfR.B1 was observed in a non-human primate (NHP) positron emission tomography (PET) study relative to control antibody. The validation using the mouse model strengthened our confidence on the predictive value of the human *in vitro* BBB model despite there is no human *in vivo* brain uptake data and limited NHP brain uptake data for biologics. Both mouse and human *in vitro* models will serve as important screening assays for brain targeted biologics selection in CNS drug development.

## Materials and methods

### Antibodies and bi-specific molecules

Sources of commercial anti-human TfR antibodies: RVS10 (Abcam ab25651), 29806 (R&D system MAB2474), OKT-9 (eBioscience 16-0719-81), MEM-189 (Novus Biologicals NB500-493), MEM-75 (Novus Biologicals NB100-77895), 3B8.2A1 (Thermo Scientific Pierce MA1-40198), ICO-92 (Thermo Scientific Pierce MA1-7657), CY1G4 (BioLegend 334102), L01.1 (BD Bioscience 347510), BGX.24 (Thermo Scientific Pierce MA1-12156), DF1513 (AbD Serotec MCA1148), Ber-T9 (Santa Cruz Biotech sc-19675). Anti-human TfR.B1 is an antibody that specifically recognizes human and monkey TfR. Anti-mouse TfR antibodies AB221, AB403, AB404, AB405 and DVD-Igs AB405-LS-RGMa and AB405-SL-RGMa are the same molecules as described before [9]. Anti-hTfR.B1 and anti-mTfR constructs were cloned into and expressed in HEK293 cells and purified according to established methods [32]. Production quantity was determined by absorbance measured with Nanodrop. Percentage of monomer was determined by size exclusion chromatography (SEC).

### Reagents

AF488 labeled human holo-transferrin (Invitrogen T13342), retinoic acid (Sigma R2625), putrescine (Sigma P5780), sodium selenite (Sigma S9133), Ro-20-1724 (Sigma B8279), apo-transferrin (Sigma T1428) were purchased from commercial sources.

### Cell culture

Human TfR overexpressing BaF3 cells were generated by transfecting human TfR1 to BaF3 cells. Primary human endothelial cells were purchased from Cell Biologics (Cat. H6023). Primary monkey brain endothelial cells were purchased from Cell Biologics (Cat. MK-6023). Primary rat astrocytes were purchased from Invitrogen (N7745-100). hCMEC/D3 cells, an immortalized brain endothelial cell line was obtained from Pierre-Olivier Couraud under license (INSERM, Paris, France). Cells were cultured according to the protocol established in the publication [21]. Caco-2 cells (ECACC) were purchased from Sigma-Aldrich (09042001). bEnd3 cells were purchased from ATCC (CRL-2299). Mouse primary brain microvascular endothelial cells were purchased from Angio Proteomie (cAP-m0002) and Cell Biologics (C57-6023). Murine intestinal epithelial cells (Cl-muADINTESTI), abbreviated as mIEC, were purchased from InSCREENeX. Monkey BBB Kit™ MBT-24H was purchased from PharmaCo-Cell. Cells were cultured and maintained as recommended by the manufacturers.

## Meso Scale Discovery (MSD)-based cell binding assay

Cells expressing transferrin receptor were added onto MSD 96-well plates (MSD Cat# L15XB-3 / L11XB-3) and incubated at 37°C for 1 hour. Cells were blocked using 15% fetal bovine serum (Hyclone, Thermo Scientific Cat# SH300700.03) at room temperature (RT) for 30 min with mild agitation; plates were then washed with Dulbecco's phosphate-buffered saline (DPBS) 3 times and 3-fold serial dilutions of antibodies from 30 μg/mL across the plate were added. After 1 hour incubation at RT, plates were washed with DPBS and goat anti-human Sulfo-TAG (MSD Cat# R32AJ-1) or anti-mouse Sulfo-Tag (MSD Cat# R32AC-1) was added. Plates were incubated at RT for 1 hour, washed with DPBS and immersed in MSD read buffer T surfactant free (MSD Cat# R92TD-2). Electrochemiluminescence (ECL) was measured on MSD SECTOR Imager 6000. EC50 values were obtained using GraphPad Prism 6 software package (GraphPad Software, Inc., La Jolla, CA).

## Fluorescence-activated cell sorting (FACS) analysis

Cells were collected in FACS buffer (1x DPBS + 2% fetal calf serum) and plated into 96-well plate at a cell density of $5 \times 10^4$ cells per well. Antibodies at indicated concentration were added to cells and incubate for 1 hour at 4°C. For detection of human TfR, phycoerythrin (PE) anti-human CD71 antibody (Biolegend Catalog number 334106) was used. After three washes with FACS buffer, cells were resuspended and analyzed with FACSCanto (BD Bioscience). For other primary antibodies without conjugation, after three washes with FACS buffer, cells were incubated for 30 minutes at 4°C with 50 μL of secondary antibody conjugated with Allophyco-cyanin (APC) diluted 1:500. After three washes with FACS buffer, cells were resuspended in 50 μL of FACS buffer and analyzed with FACSCanto (BD FACS Canto).

## Immunocytochemistry staining

Cells were plated to tissue culture plate or chamber slide overnight. Cells were treated with antibody at indicated concentration and incubated for indicated duration in cell culture medium or live cell imaging solution (Invitrogen A14291DJ). Unbound primary antibody was removed by 3 washes with PBS. Cells were fixed with 4% paraformaldehyde for 5 minutes and permeabilized with 0.3% Triton X-100 for 3 minutes. Then cells were incubated with AF594 or AF488-conjugated secondary antibody at 2 μg/mL at room temperature for 30 minutes. Then unbound secondary antibody was removed by washing 3 times with DPBS. Cell nuclei were stained with DAPI (Millipore 268298). Images were taken with fluorescence microscope or ImageXpress.

## TEER measurement

Resistance was recorded using an EVOM2 Epithelial Volt/Ohm meter coupled to a cell culture cup chamber (ENDOHM-6G) (World Precision Instruments). TEER values were presented as $\Omega \times cm^2$ subtraction of an un-seeded transwell and multiplication by 0.3 $cm^2$ to account for the surface area. TEER = $(R-R_0)$ $(\Omega) \times A(cm^2)$. TEER measurements were taken on each sample and at least in triplicate filters for each experimental condition. Measurements were made in cell culture medium.

## Transcytosis assay

Cells in 200 μL culture medium were plated on the top side of transwell (Corning #3470). One milliliter culture medium was added to the bottom side of transwell. Then cells were cultured at 37°C 5% $CO_2$ for 7–21 days before assay. 50% of medium was refreshed every 3 to 4 days

and TEER was measured with EVOM (World Precision Instruments). Antibodies were prepared as a 10x solution in cell culture medium. Then, 20 μL of antibodies was added to the upper chamber of the transwell. After indicated duration, 100 μL of sample was collected from the bottom chamber of the transwell. At the end of transcytosis assay, TEER at each transwell was measured to ensure the integrity of monolayer. Antibody concentration in these samples was determined by an ECL-MSD assay. MSD plate (MSD Cat# L15XB-3) was coated with an F (ab')$_2$ Fc fragment-specific capture antibody overnight at 4°C. Plate was blocked with 3% MSD blocking buffer (MSD Cat#R93AA-01) for 1 hour at 25°C. Plates were washed with 1X Tween-Tris buffered saline. Standards and samples diluted in 1% MSD assay buffer. Each antibody was used as an internal standard to quantify respective antibody concentrations. Plates were incubated for 2 hours at 25°C and bound antibody was detected with goat anti-human/mouse/rat Sulfo-TAG (MSD Cat# R32AH-1). Plate was read on MSD SECTOR Imager 6000. Concentration was determined from the standard curve with a five-parameter nonlinear regression program using Xlfit4 software package. N = 3 for each test article. Student's t-test compared to control IgG was performed to all transcytosis data. $p < 0.001$ (***), $p < 0.01$ (**), $p < 0.05$ (*), $p > 0,05$ (ns).

## PET imaging in non-human primates

Anti-hTfR.B1 and control IgG were conjugated, radiolabeled, and then evaluated in NHP using PET. First, antibodies were buffer exchanged to 0.1 M sodium bicarbonate pH 8.2, followed by addition of appropriate amount of p-SCN-Bn-Deferoxamine (Macrocyclics, Plano Tx) in DMSO. After incubation for 16 hrs at 23°C, the conjugated antibody was purified via protein A affinity chromatography using standard conditions. The final sample was formulated into 0.1M HEPES pH 7.1. Protein concentration was determined by BCA Protein assay (Thermo Fisher Scientific). Degree of modification was confirmed by mass spec analysis. Endotoxin levels were checked via Endosafe PTS assay (Charles River Laboratories). Aggregate content was measured by size-exclusion chromatography using a Superdex S200 column equilibrated in phosphate buffers saline, pH 7.4 (DPBS, Gibco). A cell binding assay was performed for anti-hTfR.B1 to confirm hTfR binding and to exclude impact of binding following conjugation. For radiolabeling, $^{89}$Zr in oxalic acid was diluted with 1 M HEPES pH 7.1 to below 3 mCi/mL and neutralized to pH 6.8–7.1 using 2 M NaOH and 2 M HCl. An appropriate amount of neutralized $^{89}$Zr solution was then added to the DFO-conjugated antibody to achieve molar activity of 10 mCi/mg. After mild mixing, the reaction was incubated at room temperature for 1 h. The $^{89}$Zr-labeled antibody was purified using ZEBA buffer exchange cartridge and formulated into the formulation buffer (50mM arginine, 15mM histidine, 0.02% PS80, pH 5.5). The product was passed through a sterile 0.22 μm filter and evaluated by QC (radio-TLC, SEC HPLC and endotoxin test) before injection in animals. To ensure binding capacity following radiolabeling, the immune reactive fraction (IRF) of anti-hTfR.B1 was determined to be >90%. After $^{89}$Zr labeling, the antibody was diluted to a trace concentration (<1 nM) and incubated with increasing amounts of target (expressed on Baf3-hTfR cells) in order to deplete binding-competent antibodies from solution. Unlabeled antibody at a high concentration (>1 μM) was co-incubated to determine specific binding. IRF was calculated by comparing total and bound counts in each reaction and extrapolating to infinite antigen excess using the Lindmo double-inverse method [33]. The NHP study was reviewed and approved by the Institutional Animal Care and User Committee (IACUC). For the PET studies, four protein-naïve female NHPs (*Macaca fascicularis*), ~ 8 years of age, were i.v. injected with ~1 mCi/kg (~0.1 mg/kg) of $^{89}$Zr-labeled antibody (n = 2 for anti-hTfR.B1 and n = 2 for control IgG). Animals were induced and maintained under isoflurane anesthesia for imaging on multiple

days up to 9 days post injection. Images were acquired on a MicroPET Focus 220 (Siemens Medical Solutions, Knoxville, TN) PET system and a CereTom CT (Neurologica/Samsung, Danvers, MA) scanner. PET images were reconstructed using a filtered back projection algorithm with CT-based attenuation and scatter correction. PET images were analyzed using Vivoquant software (Invicro, A Konica Minolta Company). In short, images in kBq/cc were converted to standardized uptake value (SUV), which normalizes the images by injected radioactivity and subject mass, and results in units of g/mL. Regions of interest were then generated for the brain and left ventricle of the heart using automated and manual methods. Lastly, radioactivity concentration in brain was corrected for blood volume (5%) using the radioactivity concentration in the left ventricle of the heart as an approximate of the blood radioactivity concentration.

## Western protein analysis

Cell lysates were prepared by homogenization of cell pellets in RIPA buffer (Sigma, R0278) with protease inhibitors (Sigma, 11697498001). Protein concentrations of cell lysates were determined by BCA assay (Pierce A23225). The JESS system from ProteinSimple was used to detect TfR expression in various cell lysates. Protein (1.2 µg) from each cell lysate, anti-TfR antibody (Invitrogen 13–6890, 1:200), anti-alpha-tubulin antibody (Invitrogen PA5-29444, 1:500), anti-mouse IgG (ProteinSimple 042–205) and anti-rabbit IgG (ProteinSimple 042–206) were loaded into the corresponding sections of the microplate. Protein separation in capillary cartridge by electrophoresis and immunodetection by chemiluminescence were performed in JESS automatic system. Densitometry analysis was performed using Compass software (ProteinSimple). Relative expression level of TfR was expressed as TfR normalized to tubulin.

## Results

### hCMEC/D3 model fails to discriminate transcytosis of anti-TfR antibodies from control antibody

The hCMEC/D3 model [21] is widely used for transport assay on variety of BBB targeted drug, such as small molecules with different properties [34]. Its utility on BBB targeted antibody screening remains limited [35]. This is likely due to the low TEER value [36]. As a first step towards establishing our *in vitro* BBB models, we evaluated the feasibility of using hCMEC/D3 in *in vitro* transcytosis assay. Aligned with other publications, TEER value of hCMEC/D3 model remained low regardless of the presence of astrocytes plated to the opposite side of transwell (Fig 1A) and the addition of a cocktail with supplements including retinoic acid, RO-20-1724, apo-transferrin and putrescine, [37] that might help boosting TEER. In this article, all *in vitro* transcytosis assays, except the one using commercial kits, were performed in a mono-culture setup (Fig 1B). Human TfR expression in hCMEC/D3 cells was confirmed in FACS analysis (Fig 1C). Using mono-culture setup, transcytosis of two anti-human TfR antibodies, Ber-T9 and DF1513, and mouse IgG control were performed 7-day after seeding hCMEC/D3 cells to transwells. TEER value of hCMEC/D3 monolayer reached 25 ± 2 ohm. cm$^2$ at the time of antibody treatment. Twenty nanogram (0.1 µg/mL) of anti-human TfR antibodies was added to the upper chamber of the transwells and samples were collected from lower chamber at various time point post treatment. There was minimal difference (less than 2-fold) of the amount of anti-hTfR antibodies that reached the lower chamber compared to the amount of isotype control antibody although there was significantly higher level of Ber-T9 transport at 6 and 24 hours (Fig 1D). When different amount of anti-hTfR.DF1513 antibody

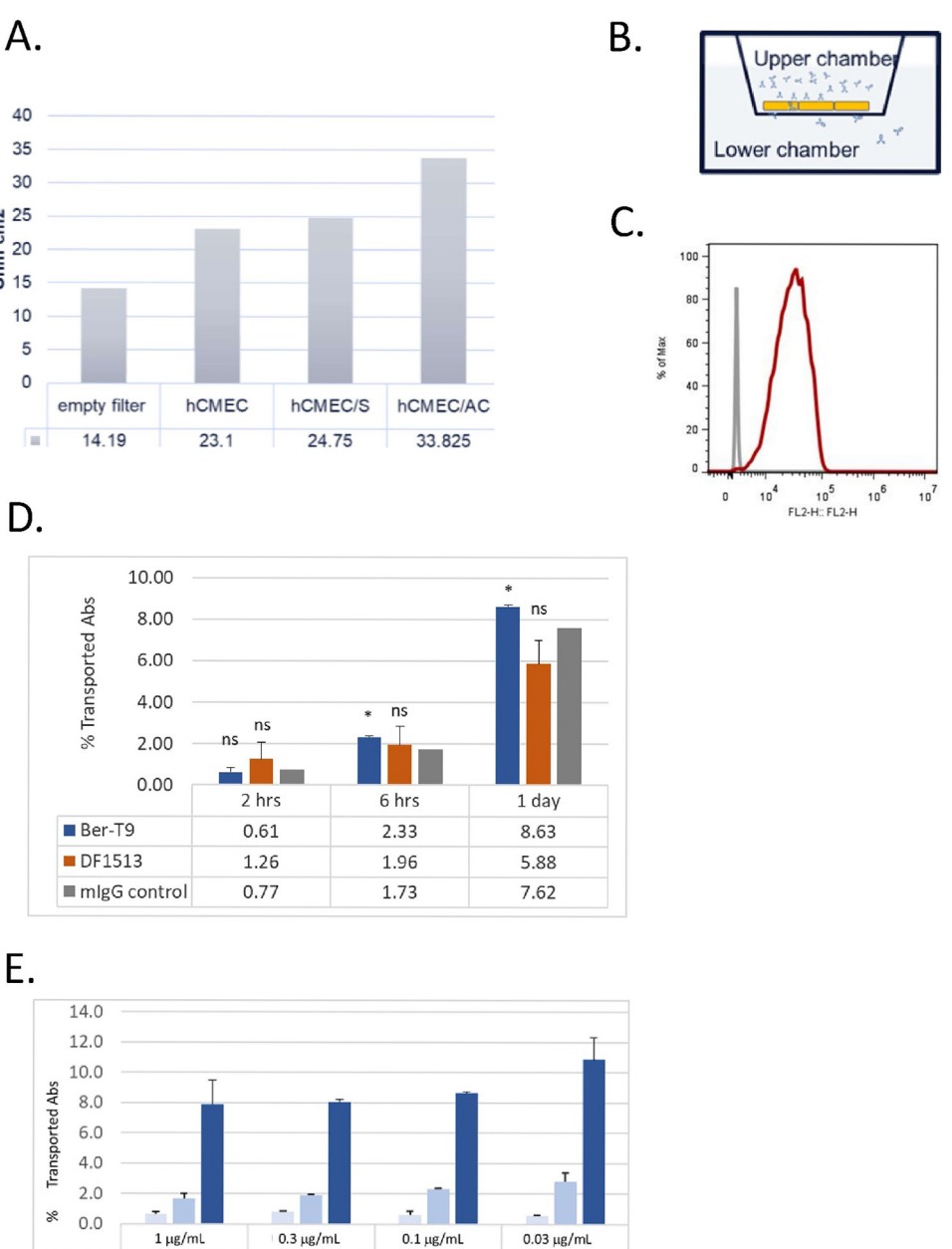

**Fig 1. hCMEC/D3 model has low TEER value and cannot discriminate transcytosis of anti-human TfR antibodies from control antibody.** (A) Integrity of hCMEC/D3-based transcytosis model in the absence or presence of astrocytes (AC) and supplements (S). TEER value at 7-day post transwell culture. (B) Scheme of antibody transcytosis assay. hCMEC/D3 cells were seeded to upper side of transwell. Antibodies were added to the upper chamber of transwell when TEER value reached plateau. Transcytosis samples were collected from the lower chamber. (C) FACS binding of anti-hTfR to hCMEC/D3 cells. (D) Transcytosis of anti-hTfR.Ber-T9 and DF1513 compared to control IgG at 2 hours, 6 hours and 1 day post 0.1 μg/mL antibody treatment. (E) Transcytosis of anti-hTfR.Ber-T9 from 0.03 μg /mL to 1 μg/ mL of antibody treatment.

was added to upper chamber, the percentile of transported anti-hTfR and control IgG is 0.3%/hour regardless of the amount of antibody input in the upper chamber (Fig 1E). The failure of the hCMEC/D3 model to discriminate transcytosis of anti-hTfR antibodies from control antibodies could be due to the low TEER value.

## Caco-2 cells form tight barriers and show various levels of anti-hTfR antibody transcytosis

To evaluate the impact of the TEER value of monolayer to paracellular transport, we tested a human intestinal epithelial cell line—Caco-2 which is one of the most extensively studied *in vitro* cell models of the human intestinal mucosa to study absorption due to its ability to form well-differentiated and polarized cell monolayers [38] that has high TEER [39]. As expected, Caco-2 cell monolayer maintained high TEER with values in the range of 2000–3000 ohm.cm$^2$ for a week after reaching its plateau (Fig 2A). Endogenous expression of human TfR in Caco-2 cells was confirmed by a FACS binding assay (Fig 2B). Anti-hTfR Ber-T9, DF1513 and IgG control were used to evaluate transcytosis of antibodies in this Caco-2 model. The transport of control IgG was 0.001% of the amount of input at 1 day post treatment (Fig 2C) which was greater than 1000-fold lower compared to the 8% transport in hCMEC/D3 model (Fig 1E). This data indicates that high TEER value is required for significant restriction of paracellular transport. Transport of anti-hTfR.Ber-T9 and anti-hTfR.DF1513 were significantly higher compared to transport of isotype control IgG (Fig 2C). The rate of antibody transport was 100 pg /cm$^2$/hour for Ber-T9; 20 ng /cm$^2$/hour for DF1513 and 8 pg /cm$^2$/hour for control IgG in Caco-2 model. To further evaluate this model, a panel of thirteen commercial anti-hTfR antibodies were screened. These antibodies demonstrated various binding potency to hTfR1 over-expressed on BaF3 cells as shown by EC$_{50}$ ranging from 0.08 nM to 10.90 nM determined by cell-based binding assay (Fig 2D). Seven out of thirteen antibodies did not show reactivity to cyno TfR. The other five antibodies showed different binding signal to cyno TfR. None of these antibodies showed binding to mTfR. These anti-hTfR antibodies demonstrated a wide range of transcytosis from 0.5–70-fold over control IgG (Fig 2D) on Caco-2 cells at 10 μg/mL. There is a trend that anti-hTfR antibodies with higher affinity (lower EC$_{50}$ value) showed no / lower level of transcytosis whereas antibodies with lower affinity (higher EC$_{50}$ value) to hTfR showed higher level of transcytosis (Fig 2D).

## A human and monkey TfR reactive anti-hTfR.B1 shows enhanced transcytosis in in vitro models and in monkey *in vivo* PET study

Anti-hTfR.B1 human IgG1 is a human and monkey TfR reactive antibody. It has binding potency of 0.2 nM (EC$_{50}$) to hTfR determined by hTfR overexpressing BaF3 cell-based binding assay and binding potency of 0.3 nM (EC$_{50}$) to cynomolgus TfR in a primary monkey cell-based FACS assay. In both Caco-2 and primary monkey brain endothelial cells, anti-hTfR.B1 internalized and colocalized with human transferrin when the cells were treated with both anti-hTfR.B1 and human transferrin for 1 hour at 37˚C. Anti-hTfR.B1 showed typical TfR-mediated endocytosis in Caco-2 cells and in primary monkey brain endothelial cells (Fig 3A). In Caco-2 cell-based transcytosis assay, the concentration-dependent curve for anti-hTfR.B1 demonstrated a linear accumulation from 0.1 μg/mL to 3 μg/mL. Transcytosis of anti-hTfR.B1 plateaued at above 3 μg/mL suggesting a saturable mechanism of transport (Fig 3B). Apparent increased transcytosis of B1 between 10 μg/mL to 100 μg/mL was in parallel with the increased transcytosis of control IgG at corresponding input concentrations, consistent with passive transport accounting for the increase.

A.

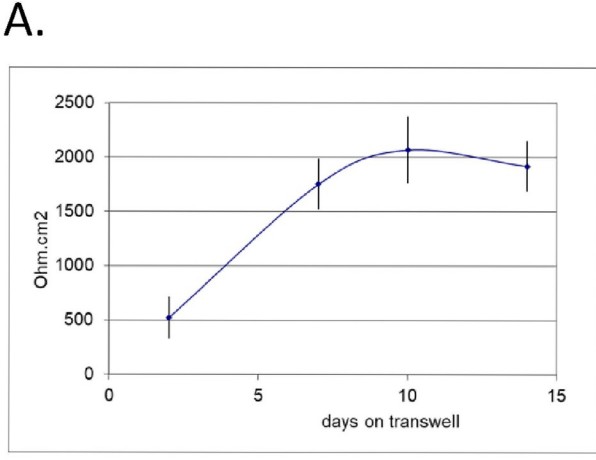

B.

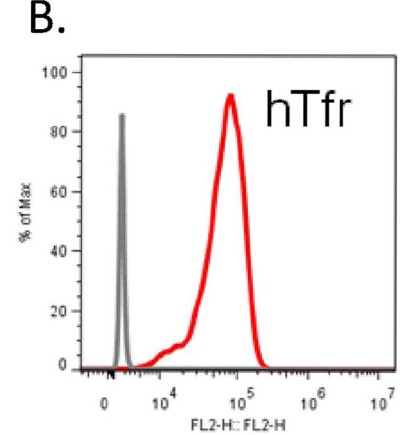

C.

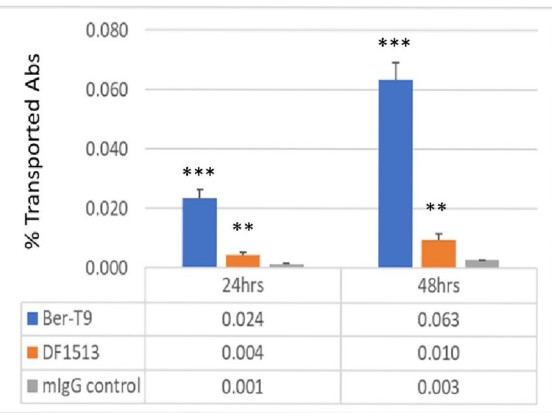

D.

| anti-hTfR1 clone name | EC50 (nM) to hTfR1 | cyno-TfR reactivity | mTfR reactivity | Caco-2 Transcytosis (fold over control IgG) | p-value transcytosis |
|---|---|---|---|---|---|
| CY1G4 | 0.08 | - | - | 3.2 | <0.001 |
| M-A712 | 0.17 | - | - | 1.3 | ns |
| MEM-75 | 0.20 | +++ | - | 7.1 | <0.001 |
| 29806 | 0.27 | - | - | 1.1 | ns |
| MEM-189 | 0.31 | - | - | 17.9 | <0.001 |
| RVS10 | 0.38 | - | - | 0.5 | ns |
| OKT-9 | 0.92 | - | - | 3.5 | <0.001 |
| DF1513 | 1.25 | +++ | - | 30.0 | <0.001 |
| Ber-T9 | 1.74 | + | - | 78.7 | <0.001 |
| 3B8 2A1 | 1.88 | - | - | 14.3 | <0.05 |
| L01.1 | 4.57 | ++ | - | 48.1 | <0.001 |
| ICO-92 | 6.73 | +/- | - | 74.7 | <0.001 |
| BGX24 | 10.90 | ++ | - | 26.4 | <0.001 |

**Fig 2. Caco-2 cells form tight barriers and show various levels of anti-hTfR antibody transcytosis.** (A) Caco-2 cells in monolayer culture reached greater than 2000 ohm.cm$^2$ resistance. (B) FACS binding of anti-hTfR to Caco-2 cells. (C) Enhanced transcytosis of anti-hTfR1.Ber-T9 and DF1513 compared to the isotype control antibody. (D) Binding and transcytosis of a panel of anti-hTfR antibodies. Binding affinity to hTfR1-BaF3 cells was performed in a cell-based MSD assay with serial dilution of antibodies. CynoTfR and mTfR reactivity were performed in a cell-based MSD assay using cynoTfR expressing cells or mTfR expressing cells at 10 µg/mL antibody concentration. Transcytosis assay in Caco-2 at 10 µg/mL input was performed. Representing data were from day 5 samples.

Next, we evaluated whether the transcytosis of anti-TfR antibodies in Caco-2 model can be replicated in brain endothelial-based model. An *in vitro* model of the BBB made of primary cultures of monkey (*Macaca irus*) brain capillary endothelial cells, brain pericytes and astrocytes called MBT BBB kit from PharmaCo-Cell [30, 40] was used. TEER value of the monkey BEC-based model was 200–500 ohm.cm$^2$. Anti-hTfR.B1 transcytosis in the MBT model was 16-fold better compared to control human IgG at 1 µg/mL and 9-fold better at 10 µg/mL (Fig 3C). Due

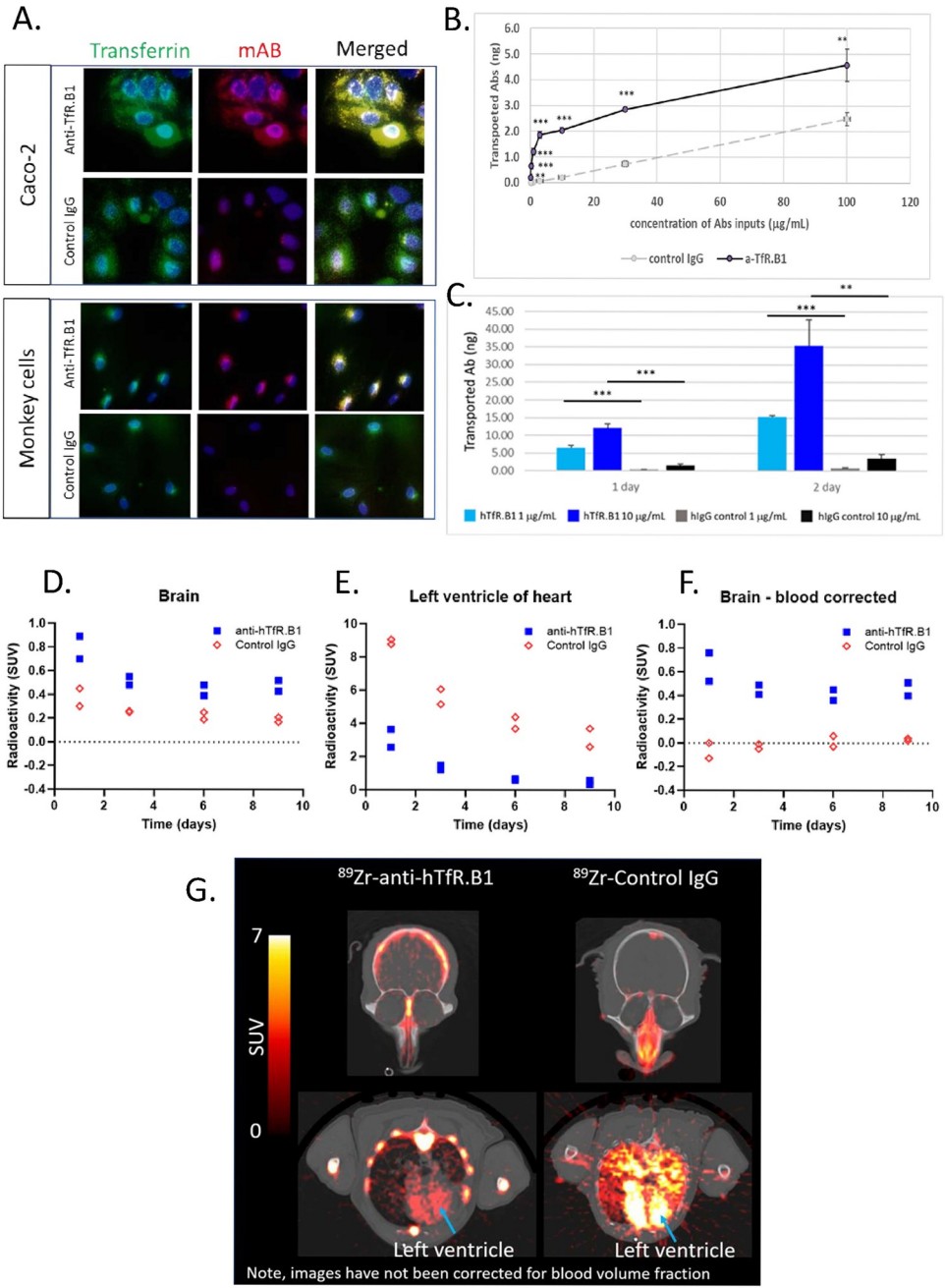

**Fig 3. The human and monkey TfR reactive anti-hTfR.B1 showed enhanced transcytosis in *in vitro* models and in *in vivo* monkey PET study.** (A) Colocalization of TfR mediated endocytosis of holo-transferrin-AF488 (Green) and anti-hTfR.B1 (Red) in Caco-2 and primary monkey brain endothelial cells. 25 μg/mL of transferrin and 10 μg/mL of antibody were mixed and added to cell culture medium for 1 hour at 37°C. Cells were fixed and stained with goat anti-human IgG-AF594 and DAPI. (B) Dose-dependent study of anti-hTfR.B1 in comparison with control IgG in Caco-2 model. N = 3 per condition, data collected from day 4. (C) Transcytosis of anti-hTfR.B1 and control IgG in primary monkey brain endothelial cell-based MBT model; (D) Anti-hTfR.B1 (n = 2) demonstrated higher brain radioactivity concentrations than control IgG (n = 2) in four NHP using PET; (E) Anti-hTfR.B1 (n = 2) demonstrated lower blood radioactivity concentrations in the left ventricle of the heart than control IgG (n = 2) in four NHPs using PET. (F) Anti-hTfR.B1 (n = 2)_demonstrated higher brain radioactivity concentrations corrected for blood contribution than control IgG (n = 2) in four NHPs using PET. (G) Representative PET/CT images which demonstrated radioactivity concentrations being higher in brain and lower in left ventricle of the heart (indicated by arrow) for anti-hTfR.B1 than control IgG as demonstrated by the axial day 1 SUV images. Antibodies were injected intravenously at 0.1 mg/kg, N = 2 for each group.

to the lack of *in vivo* brain uptake data from NHP and human, it remains unclear whether the *in vitro* transcytosis performance is correlated with its *in vivo* behavior. The fact that none of these anti-hTfR antibodies cross-react to mouse TfR excludes the possibility of using brain uptake data from mice for *in vitro in vivo* correlation. Therefore, we evaluated the brain radioactivity concentrations after injection of anti-hTfR.B1 in NHP using PET (Fig 3D–3G). PET imaging with Zirconium-89 labeled anti-hTfR.B1 demonstrated higher brain radioactivity concentrations than control Zirchonium-89 labeled IgG despite higher blood radioactivity concentrations in the heart for control Zr-89 labeled IgG. Antibodies were injected intravenously with a mass dose of 0.1 mg/kg and radioactivity dose of ~37 MBq/kg (1 mCi/kg), n = 2 for each group. Therefore, the high brain uptake of anti-hTfR.B1 relative to IgG measured with PET is consistent with the enhanced *in vitro* transcytosis of anti-hTfR.B1 compared to control IgG.

## Transcytosis of anti-mouse TfR antibodies in bEnd3 does not correlate with mouse brain uptake

Murine anti-TfR antibodies and engineered bispecific molecules provide a surrogate system to study brain shuttle performance *in vivo*. Murine brain exposure data of antibodies are available for many BBB targeted molecules. For instance, anti-mTfR antibodies and affinity variants have been demonstrated to show various levels of enhanced mouse brain uptake [9]. Brain uptake of antibodies and DVD-Igs at various doses and time points was studied in detail. Taking advantage of the available mouse brain uptake data of antibodies, development of mouse *in vitro* transcytosis model can help investigating the *in vitro in vivo* correlation.

The murine-derived bEnd3 cell line is one of the most widely used and well-characterized cell line for transport assays [41, 42]. However, in our hands, bEnd3 monolayer can only reach 50 ohm.cm$^2$ TEER at the optimal condition in which a cocktail with supplements including retinoic acid, RO-20-1724, apo-transferrin and putrescine was added. Expression of mouse TfR and endocytosis of anti-mTfR antibody in bEnd3 was confirmed by immunocytochemistry staining (Fig 4A). Anti-mTfR antibodies with high or low affinity were tested in transcytosis assay in bEnd3 cell line. AB221 is a high affinity anti-mTfR antibody (0.12 nM) that doesn't show enhanced brain exposure. AB404 and AB405 are low affinity variants (3 nM and 13.66 nM respectively) of AB221 that show 6-fold increased brain uptake compared to control antibody [9]. However, in transcytosis assay, none of these high affinity or low affinity anti-mTfR antibodies showed transcytosis higher than control IgG (Fig 4B). The transport of control IgG was around 0.2%/hour which was consistent with the data from hCMEC/D3. Other mouse models using primary mouse BEC, such as cAP-m002 from Angio-Proteomie and C57-6023 from CellBiologics, also had low TEER (16 ohm.cm$^2$ and 20 ohm.cm$^2$ respectively). Rat models using primary rat BEC [28, 29] and SV-ARBEC [27] have appreciable TEER of 100–300 ohm. cm$^2$. Some correlation with *in vivo* brain uptake for anti-rat TfR OX26 [52] and anti-IGF1R VHH mFc molecules were reported [27]. However, AB221 and its affinity variants do not bind rat TfR and so these models are unsuitable.

## mIEC model differentiates transcytosis of anti-mTfR affinity variants and shows in vitro in vivo correlation of mTfR/RGMa DVD-Ig

With success in developing *in vitro* transcytosis model with human intestinal epithelial cell line Caco-2, we started to explore an *in vitro* model with murine intestinal epithelial cells. Murine adult intestinal epithelial cells (mIEC) from InSCREENeX exhibit barrier formation function, sustaining a TEER value over 2000 ohm.cm$^2$ for a prolonged period (Fig 5A). We confirmed mIEC cells express mouse TfR by immunofluorescence (Fig 5B). Anti-mTfR antibody AB403 was rapidly internalized and retained in the puncta structure in mIEC cells which

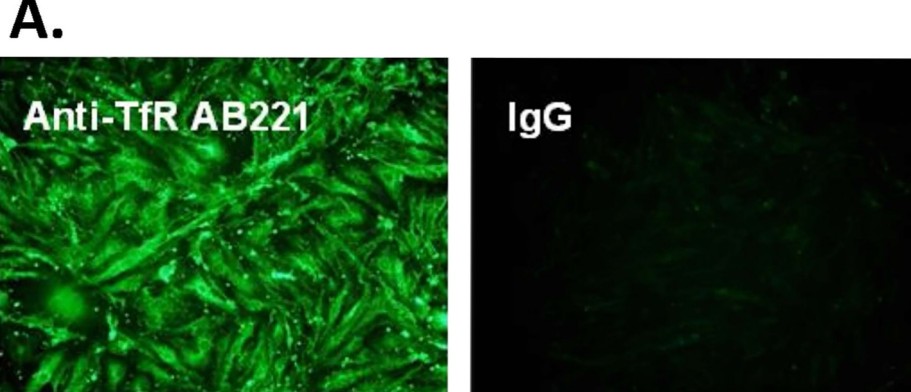

**Fig 4. Endocytosis and transcytosis of anti-mTfR antibodies in bEnd3.** (A) Immunocytochemistry staining of bEnd3 cells treated with 1 μg/ml of anti-TfR AB221 or control IgG followed by anti-human AlexaFlour-488. Binding and endocytosis of anti-mTfR AB221 to bEnd3 cells after 1 hour treatment. (B) Transport of anti-mTfR AB221, AB404 and AB405 in comparison with isotype IgG control. 1 μg/mL antibody treatment, n = 3. Binding potency of AB221, AB404 and AB405 to mTfR is 0.12 nM, 3.00 nM and 13.56 nM respectively.[9].

followed the typical TfR endocytosis pathway [43]. Binding potency of AB403 and AB405 to mIEC were 1.5 nM and 58.1 nM (EC$_{50}$) respectively in MSD binding assay. Transcytosis of control IgG was about 0.001% / hour. High affinity variant AB403 showed a similar level of transcytosis compared to control antibody in this model. The low affinity variant AB405 showed 3-fold increased transcytosis compared to control antibody (Fig 5C). These *in vitro* transcytosis data showed the same trend with the mouse brain uptake data of these antibodies (Levels of mouse brain uptake of control IgG, AB403 and AB405 are 1.1 ± 0.2 nM, 1.5 ± 0.6 nM, 7.0 ± 0.4 nM respectively 24 hours after intravenous injection) [9].

To further strengthen the *in vitro in vivo* correlation and to investigate whether the properties of TfR binding domains are retained in the context of engineered bispecific molecules, *in*

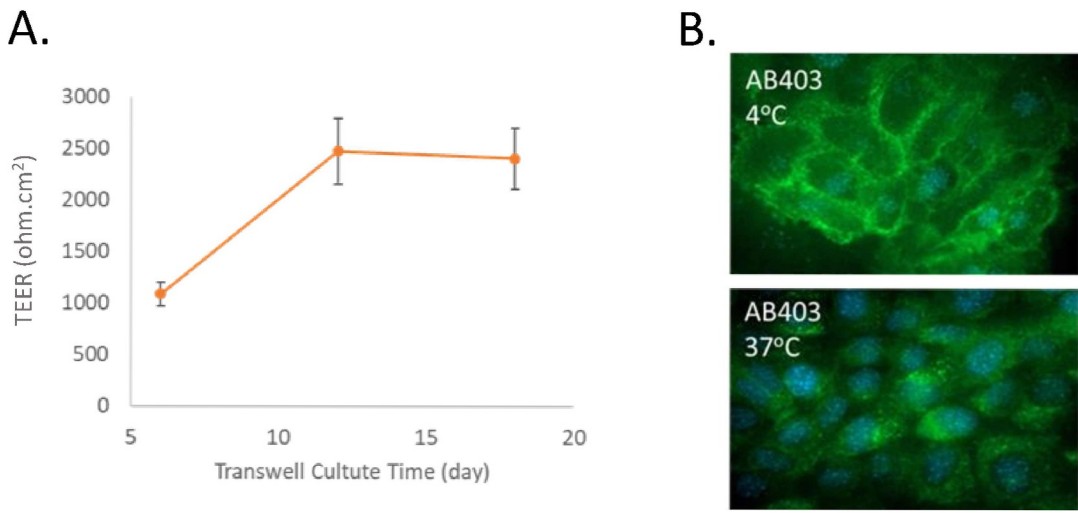

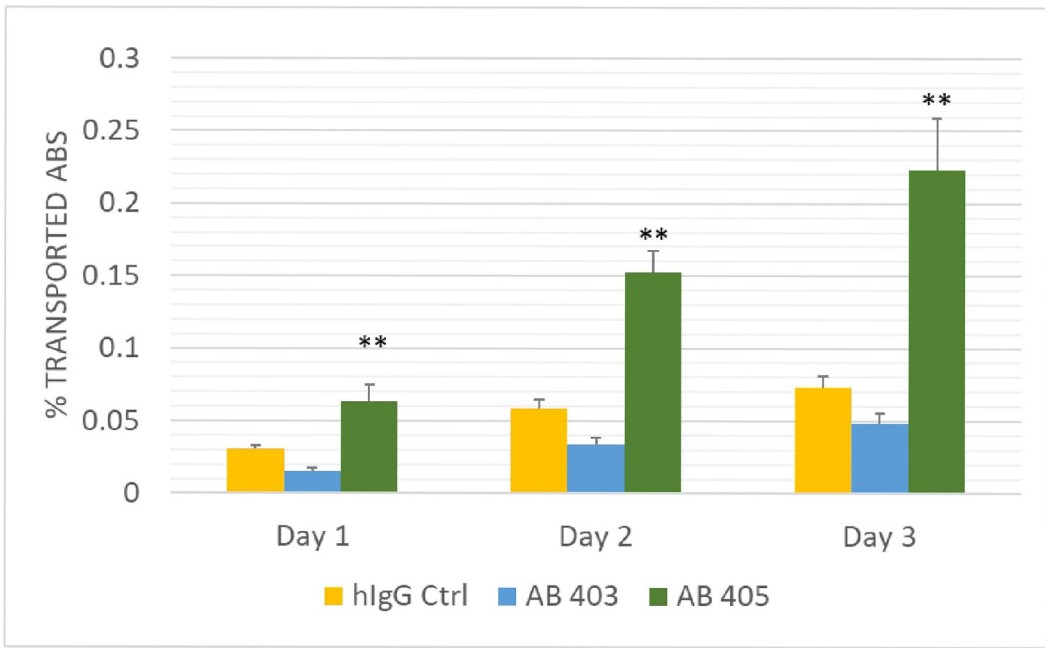

**Fig 5. mIEC monolayer forms tight barrier, expresses mTfR and differentiates transcytosis of anti-mTfR affinity variants.**
(A) mIEC cells were seeded to transwells at day 0. TEER value at various time post seeding was determined by EVOM. N = 3; (B) Anti-mTfR (20 μg/mL) incubated with mIEC cells at 4˚C or 37˚C for 1.5 hours and stained with secondary antibody conjugated with AF488. (C) Better transcytosis of low affinity variant of anti-mTfR—AB405 observed compared to high affinity variant AB403 and isotype control at 10 μg/mL input concentration.

*vitro* transcytosis of two RGMa DVD-Ig molecules using AB405 as a transport shuttle placed as the outer domain (Fig 6A) were evaluated in mIEC model. Mouse *in vivo* brain uptake of control IgG, AB403, AB405, AB405-LS-RGMa and AB405-SL-RGMa are 1.1 ± 0.2 nM, 1.5 ± 0.6 nM, 7.0 ± 0.4 nM, 18.4 ± 1.8 nM and 13.1 ± 2.8 nM respectively 24 hours after 20 mg/ kg intravenous dose [9]. mIEC transcytosis data at 72 hours post treatment of control IgG, AB403, AB405, AB405-LS-RGMa and AB405-SL-RGMa were 9.7 ± 1.0 pM, 6.5 ± 0.9 nM,

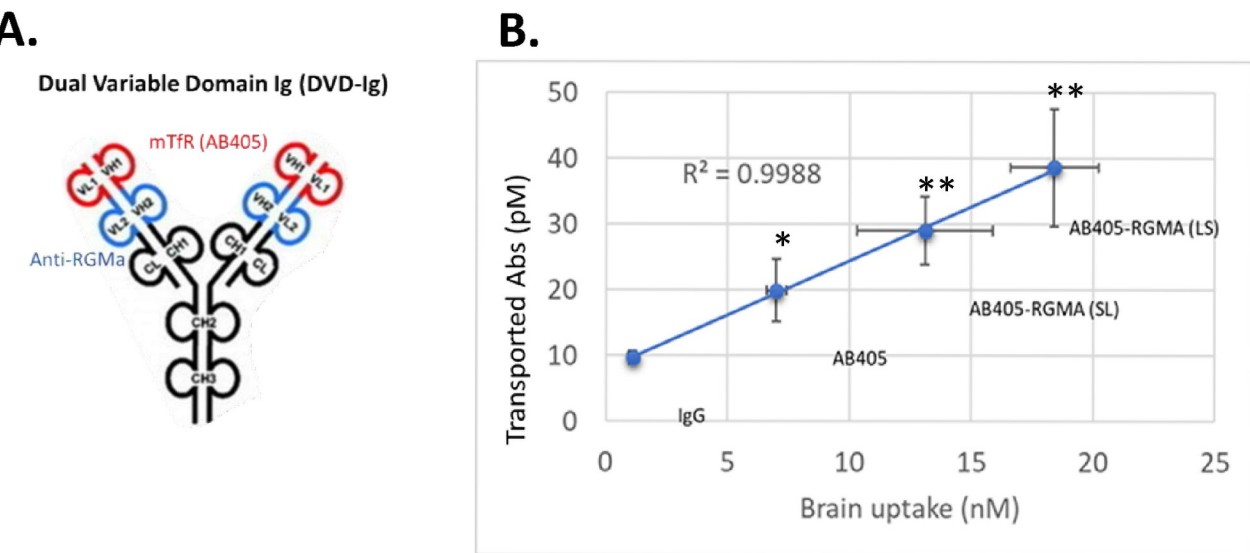

**Fig 6. *in vitro in vivo* correlation of mTfR/RGMa DVD-Ig.** (A) mTfR/RGMa DVD-Ig molecular structure graphics. LS: long-short linker; SL: short-long linker. (B) Correlation curve of mouse brain uptake data (x-axis) and mIEC *in vitro* transcytosis data (y-axis).

19.9 ± 4.7 pM, 38.6 ± 9.0 pM and 29 ± 5.1 pM respectively. *In vitro* transcytosis data from mIEC model is highly correlated with mouse *in vivo* brain exposure data (Fig 6B).

To better understand the impact of level of TfR expression on the transport of TfR targeted antibodies, TfR expression relative to tubulin was determined by Western protein analysis using cell lysates from cells that were used in transcytosis model and primary human BEC, monkey BEC and mouse BEC. All cells expressed TfR. Level of TfR expression ranged from 1 to 2.5-fold over tubulin except bEnd3 cells (0.7-fold) (Fig 7).

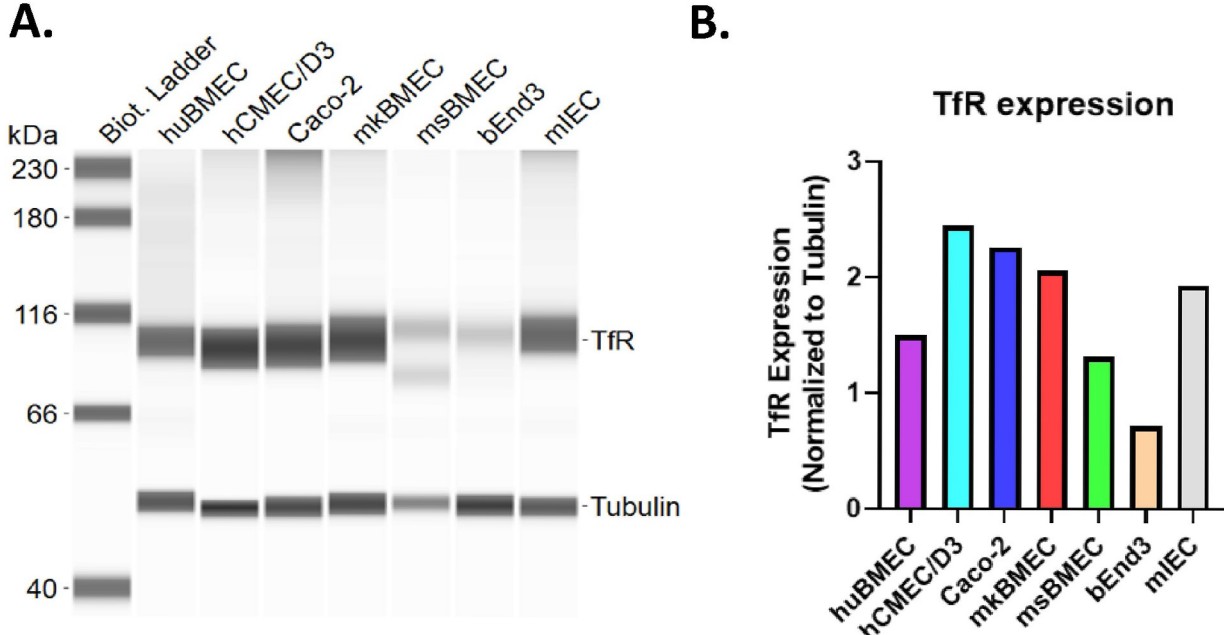

**Fig 7. TfR protein expression in primary brain endothelial cells and cells used in *in vitro* transcytosis model.** (A) Western protein analysis of TfR expression in the cells used in *in vitro* transcytosis models and primary brain endothelial cells. (B) Quantitative analysis of the level of TfR expression relative to tubulin.

## Discussion

In this study, we evaluated multiple *in vitro* transcytosis models for their potential application for CNS targeted biologics drug screening. By comparing transcytosis of anti-hTfR antibody Ber-T9, DF1513 and control IgG in hCMEC/D3 and Caco-2, we found out the importance of high TEER value in transcytosis assays. This phenomenon is also observed in the murine models by comparing the transcytosis of anti-mTfR molecules in bEnd3 and mIEC models. Models with TEER value below 100 ohm.cm$^2$, such as hCMEC/D3 and bEnd3, show high level of paracellular transport of antibody which masks the transcytosis of anti-TfR through the RMT route. The monkey model with TEER value between 200 to 500 ohm.cm$^2$ showed good differentiation between TfR-targeted antibody and control IgG indicating a model threshold of a few hundred ohm.cm$^2$ tightness might be sufficient to block antibody diffusion. Consistent with this notion, it has been reported that a rat RBE model with 100–300 ohm.cm$^2$ TEER value can differentiate the transcytosis of anti-IGF1R molecules [27]. The low TEER threshold of a good *in vitro* transcytosis model likely depends on the expression and transcytosis efficiency of the BBB target and the dynamic interactions between the BBB-targeted molecule and its receptor. The low TEER limit also depends on the sensitivity of the detection method. Improvement on the sensitivity of biologics detection that enables screening at low concentration can help improve the signal (RMT transcytosis) to noise (paracellular transport) ratio. Models, such as Caco-2 and mIEC, have greater than 1000 ohm.cm$^2$ TEER value and thus result in extremely low level of paracellular transport of biologics which makes these models suitable for the screening of molecules that pass through the cells in RMT pathway. The fact that these models can maintain at high TEER value for over a week allows the accumulation of transported molecules at the basolateral site so that it can be detected by our assay with 0.1 ng/mL low limit of quantitation (LLOQ).

For anti-hTfR.B1, its maximal transcytosis rate in Caco-2 and in MBT is 0.4 nmol/cm$^2$/h and 13 nmol/cm$^2$/h respectively which was calculated by the amount of transported anti-hTfR. B1 subtracted the amount of transported control IgG at 10 μg/ml input condition at day 2. The relative level of TfR expression in Caco-2 and primary monkey cells is comparable (Fig 7). This data indicates a much more rapid RMT transcytosis in brain endothelial cells compared to gut cells. Dose-response study of anti-hTfR.B1 indicates that there is a maximal transport capacity of a specific antibody / receptor combination. For anti-hTfR.B1, the best dose is 3 μg/mL which is close to the concentration that shows maximal binding to hTfR expressing cells. Optimal *in vitro* transcytosis (Fig 2D) and optimal brain uptake [44] of TfR targeted molecule can be modulated by altering its affinity to TfR [8, 9]. The transcytosis of anti-hTfR.B1 in brain endothelial cells is about 15-fold and 8-fold higher compared to control IgG at 1μg/mL and 10 μg/mL respectively. Dense brain capillary network offers large area for transport in human brain. The average adult human brain weighs about 1500 g. The surface area of microvessels is 100–200 cm$^2$ g$^{-1}$ tissue [45, 46], corresponding to a total surface area of 15–25 m$^2$ [47]. Although the architecture of the brain microvasculature is very similar across species [47], the expression levels of transporters [48] and RMT target varied [49]. Due to the large BBB area, it is feasible to transport therapeutically meaningful number of biologics to brain through RMT route.

Albeit Caco-2 and mIEC are epithelial cells, and BBB is composed of endothelial cells, they share a common function of lining lumens / cavities, they have a protective function mediated in part by tight junctions and they share a property of having oriented transport functions. The best-known transcytosis events that happen physiologically are caveolae-mediated transcytosis by endothelial cells and clathrin-mediated transcytosis of IgA by epithelial cells [50]. However, these studies are focused on transport of macromolecules that do not bind to a cell

membrane target. In contrast, TfR is a ubiquitously expressed cell membrane receptor that endocytose through clathrin-mediated endocytosis pathway [43]. Our data indicated that the endocytosis of anti-TfR in Caco-2 and mIEC follows a typical endocytic pattern of TfR. We showed that good transcytosis of anti-hTfR.B1 in Caco-2 can be replicated in primary monkey brain endothelial cell-based multi-cellular model. More importantly, in a monkey PET study intravenously injected anti-hTfR.B1 enriched in brain. In addition, *in vitro* transcytosis of anti-mTfR antibodies in mIEC model correlates well with their *in vivo* brain uptake in mice. Collectively, this implies that despite the differences in physiological function and structure, endothelial cells and epithelial cells might share similar RMT transport mechanisms. Gut epithelial cells might not be reflecting absolute brain transport rate of TfR targeted molecules and thus their main utility is in comparative studies such as affinity optimization of TfR targeted antibody, and *in vitro in vivo* correlative studies.

The determinants of different pathways of intracellular trafficking of BBB-targeted biologics remain largely unknown. In some cases, binding affinity and valency determined the sorting to different pathways [51]. Endosomal trafficking [52] or $_p$H-dependent dissociation of antibody and targeted receptor [35] regulates receptor-mediated transcytosis of antibodies across the blood brain barrier. We screened transcytosis of a small panel of anti-TfR antibodies (Fig 2D) with various affinities to hTfR; in general, lower affinity clones tend to have higher level of transcytosis. However, a few clones, such as MEM-189, 3B8.2A1 and BGX,24, are outliners which indicate that there are other factors such as binding epitope, charge, and poly-specificity in addition to binding affinity that impact the transcytosis of antibodies. There remains much to be learned about the sorting and regulation of the transcellular transport through the endosomal system in cellular models. Establishing *in vitro* models that show relevance to *in vivo* brain uptake of molecules also helps in selecting models for these important mechanistic studies.

*In vivo* brain uptake of biologics depends on multiple factors in addition to BBB crossing, including phenomena that influence systemic exposure such as receptor mediated disposition and serum stability. *In vitro* models are only focused on the transport of biologics at brain endothelial cells and are thus inherently oversimplified models. To fully predict the *in vivo* brain penetration of biologics, additional assays to investigate other related parameters are needed to couple with the *in vitro* transcytosis data. Also, different BBB targets, such as CD98hc, InsR and IGF1R [14, 27, 53], are likely to employ different transcytosis mechanisms. While the gut cell-based models are useful for common broadly expressed targets such as TfR, they will be less useful for brain endothelial specific targets. The potential application of these models to other BBB targets needs to be evaluated in a case-by-case scenario. An ideal transcytosis model with broad applications should not only have a high TEER value but also be easy to transfect to express receptor-of-interest and have highly reproducible performance at high passage numbers. Development of higher sensitivity antibody detection methods can enable high throughput screening of candidates and it can also enable transcytosis assay with trace amount of antibody which will significantly minimize paracellular transport.

## Conclusions

Transcytosis of anti-TfR targeted antibodies were evaluated in multiple *in vitro* transcytosis models. High TEER value is a key parameter to differentiate RMT-mediated transcytosis from non-specific paracellular transport of antibodies. Our data showed good differentiation of transcytosis of a panel of anti-hTfR antibodies in Caco-2 and we confirmed good transcytosis of anti-hTfR.B1 in MBT model and monkey *in vivo* PET study. The validation using the mouse model strengthened our confidence on the predictive value of the human *in vitro* BBB

model despite there is no human *in vivo* brain uptake data and limited NHP brain uptake data for biologics. Both mouse and human *in vitro* transcytosis models will serve as important screening assays for BBB targeted antibody / bispecific selection.

## Supporting information

**S1 Data.**
(DOCX)

## Acknowledgments

The authors are thankful to AbbVie employees John E. Harlan, Ji-Quan Wang, Kyle C. Wilcox, David Reuter, Martin J. Voorbach and Dustin Wooten for their expert contributions to the PET study and thank former AbbVie employee Farid Gizatullin for his help with MSD-ELISA assay. We are also thankful to AbbVie employee Axel Meyer and former AbbVie employee Denise Karaoglu Hanzatian for their support on hCMEC/D3 model.

## Author Contributions

**Conceptualization:** Kangwen Deng.

**Data curation:** Kangwen Deng, Yifeng Lu, Sjoerd J. Finnema, Kostika Vangjeli, Junwei Huang.

**Formal analysis:** Kangwen Deng, Yifeng Lu, Sjoerd J. Finnema.

**Investigation:** Sjoerd J. Finnema, Kostika Vangjeli.

**Resources:** Andrew Goodearl.

**Supervision:** Kangwen Deng, Andrew Goodearl.

**Visualization:** Kangwen Deng, Yifeng Lu, Sjoerd J. Finnema, Junwei Huang.

**Writing – original draft:** Kangwen Deng, Yifeng Lu, Sjoerd J. Finnema, Lili Huang, Andrew Goodearl.

**Writing – review & editing:** Kangwen Deng, Yifeng Lu, Sjoerd J. Finnema, Lili Huang, Andrew Goodearl.

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
