## [Decision Letter · Decision Letter 0]

10 May 2023

PONE-D-23-10964Application of In vitro Transcytosis Models to Brain Targeted BiologicsPLOS ONE

Dear Dr. Deng,

Thank you for submitting your manuscript to PLOS ONE. After careful consideration, we feel that it has merit but does not fully meet PLOS ONE’s publication criteria as it currently stands. Therefore, we invite you to submit a revised version of the manuscript that addresses all the points raised during the review process.

We look forward to receiving your revised manuscript.

Kind regards,

Mária A. Deli, M.D., Ph.D.

Academic Editor

PLOS ONE

Journal Requirements:

   "This study was sponsored by AbbVie. AbbVie participated in design and conduct of experiments, interpretation of data, and review and approval of the manuscript. "

Additional Editor Comments:

Two experts have evaluated the manuscript and agreed that it has merits but need to be further improved. The most important comments to be addressed are as follows: 1. Both reviewers ask for statistical analyses of the data in all figures.2. The correction of the brain levels of the antibodies to the heart levels seems to  introduce bias in the data and the uncorrected values should be also shown. 3. The presence of  TfR either at mRNA or protein levels should be verified on the in vitro models. 

Reviewers' comments:

Reviewer's Responses to Questions

**Comments to the Author**

1. Is the manuscript technically sound, and do the data support the conclusions?

Reviewer #1: Yes

Reviewer #2: Yes

2. Has the statistical analysis been performed appropriately and rigorously? 

Reviewer #1: No

Reviewer #2: No

3. Have the authors made all data underlying the findings in their manuscript fully available?

Reviewer #1: Yes

Reviewer #2: Yes

4. Is the manuscript presented in an intelligible fashion and written in standard English?

Reviewer #1: Yes

Reviewer #2: Yes

5. Review Comments to the Author

Reviewer #1: The authors evaluated transcytosis of anti-TfR targeted antibodies in multiple in vitro transcytosis models. The authors suggest that high TEER value is a key parameter to differentiate RMT-mediated transcytosis from non-specific, paracellular transport of antibodies. The author also evaluated transcytosis of a panel of anti-hTfR antibodies in Caco-2 and MBT model. They also evaluated one of the antibodies in monkey in vivo PET study. Finally, they evaluated anti-TfR antibodies in a mouse model. Below are my comments/questions:

1. The authors need to do stats on all their findings to claim any significance or non significance.

2. In Figure 3, the number of replicates is only 2 for some of the panels. Please confirm as stats cannot be done on these findings.

3. Throughout the text and figures, the authors mention the terms ‘brain concentration’ and ‘blood concentration’ several times especially for PET imaging. However, concentrations such as values in pg/mL or nM are not provided. If the authors have these values, please provide them. Otherwise, the correct the terms to ‘relative levels’.

4. In the results for PET imaging, the authors claim that the control IgG has higher blood concentrations than anti-hTfR.B1. Could this be due to different stability of the two antibodies in vivo? Has the PK of the two antibodies been compared in rodents?

5. Why was the brain concentration corrected against the heart concentration (as is mentioned in Materials and Methods for PET imaging)? Since the heart levels for anti-hTfR.B1 is lower than the control IgG, correcting the brain levels of the two antibodies against the heart levels would always give a higher value for anti-hTfR.B1. That is, the results shown in Fig 3D could be an artifact of the heart correction (Fig 3E) and the actual brain levels of anti-hTfR.B1 might not be that different than control IgG. To avoid this confusion, can the authors provide the uncorrected values for brain levels?

6. In Fig 6, the authors show in vitro and in vivo correlation of various anti-mTfR antibodies. For the brain concentration (in vivo), do the authors have evidence that the concentration is mostly in the brain parenchyma and not brain vessels?

7. The hCMEC/D3 model failed to discriminate transcytosis of anti-hTfR vs control antibodies. Can the authors confirm if their cells still express the hTfR?

Reviewer #2: This manuscript submitted by Kangwen Deng et al. evaluated transcytosis of anti-human transferrin receptor (TfR) antibodies using several in vitro models derived from human, cynomolgus, and mouse species. The authors found that models with low barrier properties (low TEER), including hCMEC/D3 and bEnd3, were not suitable for in vitro transcytosis assays. Meanwhile, the authors indicated that Caco-2, a monkey brain endothelial cell-based model, and mIEC, which have high barrier properties, were appropriate as in vitro transcytosis models.

Globally, the study appears to be well performed, and the results are interesting. Thus, some points need to be addressed by the authors to improve the quality of the article.

1. Statistics

The authors should discuss this based on the statistical results, even if they are obvious.

Please perform statistical analysis and describe the statistical methods and results.

2. Figure 3F.

Please show a PET image of the heart for comparison with that of the brain.

3. Comparative data on TfR protein expression levels in each model might be useful for understanding the differences in RMT transcytosis between models. Did the author examine expression levels in each model?

4. Discussion section (p17 lane 12)

For anti-hTfR.B1, its maximal transcytosis rate in Caco-2 and in MBT is 0.4 nmol/cm2/h and 13 nmol/cm2/h respectively…..

Please also show the data as the fold-over IgG controls. The MBT model might have higher permeability through the paracellular route than the Caco-2 model because of the difference in tight junction functions (TEER value).

5. Please check the following points.

1) Figure 5A, Y-axis: ohm/cm2 => ohm x cm2

2) Discussion, p17 Lane 16: Optimal in vitro transcytosis (Figure 4C) …=> I could not find Figure 4C

3) Figure 3 legend, p23: …NHP using PE€(E) => PET

6. PLOS authors have the option to publish the peer review history of their article (what does this mean?). If published, this will include your full peer review and any attached files.

Reviewer #1: No

Reviewer #2: No

---

## [Author Response · Author response to Decision Letter 0]

12 Jul 2023

5. Review Comments to the Author

Reviewer #1: The authors evaluated transcytosis of anti-TfR targeted antibodies in multiple in vitro transcytosis models. The authors suggest that high TEER value is a key parameter to differentiate RMT-mediated transcytosis from non-specific, paracellular transport of antibodies. The author also evaluated transcytosis of a panel of anti-hTfR antibodies in Caco-2 and MBT model. They also evaluated one of the antibodies in monkey in vivo PET study. Finally, they evaluated anti-TfR antibodies in a mouse model. Below are my comments/questions:

1. The authors need to do stats on all their findings to claim any significance or non significance. 

The stats were added to all data.

2. In Figure 3, the number of replicates is only 2 for some of the panels. Please confirm as stats cannot be done on these findings. 

PET studies were indeed performed in two NHPs for each radiolabeled antibody. This has now been clarified in the legend of Figure 3. We agree with the reviewer that it is not appropriate to perform statistical analysis on a dataset with this limited number of subjects. Protein-naïve NHPs are very challenging to access and studies are typically restricted to a limited sample size. However, data from PET measurements in NHPs has been shown highly reproducible and provide important translational value. 

3. Throughout the text and figures, the authors mention the terms ‘brain concentration’ and ‘blood concentration’ several times especially for PET imaging. However, concentrations such as values in pg/mL or nM are not provided. If the authors have these values, please provide them. Otherwise, the correct the terms to ‘relative levels’.

PET is used to measure the amount of radioactivity per volume of blood (kBq/mL) or tissue (kBq/cc). We therefore prefer to use the term radioactivity concentration. The radioactivity concentration is expressed in the standardized uptake value (SUV) unit. SUV is a commonly used unit and includes normalization for the amount of injected radioactivity and the subject mass and thereby allows for direct comparison between subjects and across species. For further clarification, we have now updated relevant text to ‘radioactivity concentration’ throughout the manuscript.

4. In the results for PET imaging, the authors claim that the control IgG has higher blood concentrations than anti-hTfR.B1. Could this be due to different stability of the two antibodies in vivo? Has the PK of the two antibodies been compared in rodents? 

There was no indication that the radiolabeled antibodies were unstable in vivo. The two antibodies were also conjugated and radiolabeled using the same conditions and methods. We did not compare the PK of these two antibodies in rodents. Anti-TfR antibodies are known to have fast clearance due to target-mediated drug disposition (TMDD) based on our internal data and literature. Anti-hTfR.B1 binds cyno TfR but not rodent TfR, thus is expected to exhibit fast clearance in cyno monkey but not in rodents due to TMDD. The control antibody has a longer half-life in blood (which is consistent with other experiments not included in this manuscript). 

5. Why was the brain concentration corrected against the heart concentration (as is mentioned in Materials and Methods for PET imaging)? Since the heart levels for anti-hTfR.B1 is lower than the control IgG, correcting the brain levels of the two antibodies against the heart levels would always give a higher value for anti-hTfR.B1. That is, the results shown in Fig 3D could be an artifact of the heart correction (Fig 3E) and the actual brain levels of anti-hTfR.B1 might not be that different than control IgG. To avoid this confusion, can the authors provide the uncorrected values for brain levels? 

In PET measurements of the brain, it is common practice to correct for the blood volume present in the brain. About 5% of the total brain volume represents blood. Since radioactivity concentration in the blood is considerable, the correction is performed to obtain an accurate estimation of the radioactivity concentration in the brain tissue. Figure 3F was added to include the brain radioactivity concentrations which were not corrected for blood contribution. Figure 3G was updated and now includes PET images of the brain and the heart. PET images were not corrected for blood volume. It can be observed in Figure 3F and 3G that the radioactivity concentration in brain is higher for anti-hTfR-B1 than control IgG, even when no blood volume correction is applied for the brain.

6. In Fig 6, the authors show in vitro and in vivo correlation of various anti-mTfR antibodies. For the brain concentration (in vivo), do the authors have evidence that the concentration is mostly in the brain parenchyma and not brain vessels? In Karaoglu 2018 paper (Reference #9), immunohistology images of anti-mTfR and DVD-Igs containing anti-mTfR domain show that the concentration is not only from vasculature but also from molecules that penetrated brain parenchyma.

7. The hCMEC/D3 model failed to discriminate transcytosis of anti-hTfR vs control antibodies. Can the authors confirm if their cells still express the hTfR? Yes, hCMEC/D3 expresses human TfR. FACS data added (Figure 1C).

Reviewer #2: This manuscript submitted by Kangwen Deng et al. evaluated transcytosis of anti-human transferrin receptor (TfR) antibodies using several in vitro models derived from human, cynomolgus, and mouse species. The authors found that models with low barrier properties (low TEER), including hCMEC/D3 and bEnd3, were not suitable for in vitro transcytosis assays. Meanwhile, the authors indicated that Caco-2, a monkey brain endothelial cell-based model, and mIEC, which have high barrier properties, were appropriate as in vitro transcytosis models.

Globally, the study appears to be well performed, and the results are interesting. Thus, some points need to be addressed by the authors to improve the quality of the article.

1. Statistics 

The authors should discuss this based on the statistical results, even if they are obvious.

Please perform statistical analysis and describe the statistical methods and results.

The Stats were added to all data

2. Figure 3F.

Please show a PET image of the heart for comparison with that of the brain. Figure 3G was updated and now includes PET images of the brain and the heart.

3. Comparative data on TfR protein expression levels in each model might be useful for understanding the differences in RMT transcytosis between models. Did the author examine expression levels in each model? Figure 7 - TfR protein expression in primary brain endothelial cells and cells that were used in in vitro transcytosis model, was added.

4. Discussion section (p17 lane 12)

For anti-hTfR.B1, its maximal transcytosis rate in Caco-2 and in MBT is 0.4 nmol/cm2/h and 13 nmol/cm2/h respectively…..

Please also show the data as the fold-over IgG controls. The MBT model might have higher permeability through the paracellular route than the Caco-2 model because of the difference in tight junction functions (TEER value). 

Background paracellular route transport has been subtracted from this calculation. The fold-over IgG controls vary at different input conditions. For instance, referring to the data in Figure 3B, the fold over IgG at 1, 3, 10, 30 ug/ml is 93x, 24x, 10x and 4x respectively. The rate of anti-TfR.B1 transport remains the same when anti-TfR.B1 input is greater than 3 ug/ml. A sentence describing how the data was obtained was added to the text.

5. Please check the following points. 

1) Figure 5A, Y-axis: ohm/cm2 => ohm x cm2 Figure was corrected.

2) Discussion, p17 Lane 16: Optimal in vitro transcytosis (Figure 4C) …=> I could not find Figure 4C That should be Figure 2D. The text was corrected.

3) Figure 3 legend, p23: …NHP using PE€(E) => PET The text was corrected.

---

## [Decision Letter · Decision Letter 1]

31 Jul 2023

Application of In vitro transcytosis models to brain targeted biologics

PONE-D-23-10964R1

Dear Dr. Deng,

We’re pleased to inform you that your manuscript has been judged scientifically suitable for publication and will be formally accepted for publication once it meets all outstanding technical requirements.

Kind regards,

Mária A. Deli, M.D., Ph.D.

Academic Editor

PLOS ONE

Additional Editor Comments (optional):

Reviewers' comments:

Reviewer's Responses to Questions

**Comments to the Author**

1. If the authors have adequately addressed your comments raised in a previous round of review and you feel that this manuscript is now acceptable for publication, you may indicate that here to bypass the “Comments to the Author” section, enter your conflict of interest statement in the “Confidential to Editor” section, and submit your "Accept" recommendation.

Reviewer #1: All comments have been addressed

Reviewer #2: (No Response)

2. Is the manuscript technically sound, and do the data support the conclusions?

Reviewer #1: Yes

Reviewer #2: (No Response)

3. Has the statistical analysis been performed appropriately and rigorously? 

Reviewer #1: Yes

Reviewer #2: (No Response)

4. Have the authors made all data underlying the findings in their manuscript fully available?

Reviewer #1: Yes

Reviewer #2: (No Response)

5. Is the manuscript presented in an intelligible fashion and written in standard English?

Reviewer #1: Yes

Reviewer #2: (No Response)

6. Review Comments to the Author

Reviewer #1: (No Response)

Reviewer #2: There are very small points.

1) It seems that the Y-axis in Fig5A has not yet been corrected.

2) Please adjust the position of Fig5 legend.

All other questions raised by the reviewer were addressed. It is clear and well-considered.

7. PLOS authors have the option to publish the peer review history of their article (what does this mean?). If published, this will include your full peer review and any attached files.

Reviewer #1: No

Reviewer #2: No

---

## [Editor Report · Acceptance letter]

8 Aug 2023

PONE-D-23-10964R1 

Application of *In vitro* transcytosis models to brain targeted biologics 

Dear Dr. Deng:

I'm pleased to inform you that your manuscript has been deemed suitable for publication in PLOS ONE. Congratulations! Your manuscript is now with our production department. 

Kind regards, 

on behalf of

Prof. Mária A. Deli 

Academic Editor

PLOS ONE